# Oxygen Uptake and Bilaterally Measured Vastus Lateralis Muscle Oxygen Desaturation Kinetics in Well-Trained Endurance Cyclists

**DOI:** 10.3390/jfmk8020064

**Published:** 2023-05-13

**Authors:** Karmen Reinpõld, Indrek Rannama

**Affiliations:** School of Natural Sciences and Health, University of Tallinn, 10120 Tallinn, Estonia; karmen.reinpold@tlu.ee

**Keywords:** V̇O_2_ kinetics, NIRS kinetics, V̇O_2_ slow component, V̇O_2_ fast component, Moxy Monitor

## Abstract

The aim of the present study was to compare and analyse the relationships between pulmonary oxygen uptake and vastus lateralis (VL) muscle oxygen desaturation kinetics measured bilaterally with Moxy NIRS sensors in trained endurance athletes. To this end, 18 trained athletes (age: 42.4 ± 7.2 years, height: 1.837 ± 0.053 m, body mass: 82.4 ± 5.7 kg) visited the laboratory on two consecutive days. On the first day, an incremental test was performed to determine the power values for the gas exchange threshold, the ventilatory threshold (VT), and V̇O_2max_ levels from pulmonary ventilation. On the second day, the athletes performed a constant work rate (CWR) test at the power corresponding to the VT. During the CWR test, the pulmonary ventilation characteristics, left and right VL muscle O_2_ desaturation (DeSmO_2_), and pedalling power were continuously recorded, and the average signal of both legs’ DeSmO_2_ was computed. Statistical significance was set at *p ≤* 0.05. The relative response amplitudes of the primary and slow components of VL desaturation and pulmonary oxygen uptake kinetics did not differ, and the primary amplitude of muscle desaturation kinetics was strongly associated with the initial response rate of oxygen uptake. Compared with pulmonary O_2_ kinetics, the primary response time of the muscle desaturation kinetics was shorter, and the slow component started earlier. There was good agreement between the time delays of the slow components describing global and local metabolic processes. Nevertheless, there was a low level of agreement between contralateral desaturation kinetic variables. The averaged DeSmO_2_ signal of the two sides of the body represented the oxygen kinetics more precisely than the right- or left-leg signals separately.

## 1. Introduction

It is well-established knowledge that endurance performance is highly correlated with parameters such as maximal oxygen uptake (V̇O_2max_), exercise economy, the lactate threshold, critical power (CP), and performance O_2_ deficit [1]. Nevertheless, they are not the parameters to “determine” the performance outcome but rather characterise the kinetics of oxygen uptake (V̇O_2_), which itself determines aerobic and anaerobic energy processes in the body and therefore contributes to the performance outcome [2,3].

During exercise, oxygen kinetics can be divided into different phases. Phase I is a cardiodynamic component, where increased pulmonary blood flow requires increased cardiac output, resulting in an increase in oxygen uptake (V̇O_2_). Phase II is the primary component, where V̇O_2_ rises rapidly as oxygen needs to increase in the working muscle. Phase III is the steady-state phase but can develop into the slow component, where V̇O_2_ keeps increasing depending on exercise intensity [4,5]. The V̇O_2_ slow component represents a slowly developing increase in V̇O_2_ during a steady-state workload performed above the lactate threshold [2]. It is considered a central component to exercise tolerance [6] and sports performance. Many different factors are proposed to be contributing to the V̇O_2_ slow component. Nevertheless, it is primarily attributed to the progressive recruitment of type II muscle fibres [7]. Sprinters are more prone to a higher slow component than endurance athletes [8], but specific endurance training adaptation can increase time to exhaustion and raise endurance performance [9].

V̇O_2_ kinetics is positively correlated with V̇O_2max_ [3], and four exercise intensity domains are traditionally described: moderate, heavy, severe, and extreme [5]. In the moderate domain, a steady state is reached quickly [10]. In the heavy domain, there is a slow component and delayed steady state [5]. In the severe domain, the slow component does not stabilise until maximum oxygen uptake is reached [6]. The border between heavy and severe is unclear, and some authors [11] use additional classifications. Most studies use the relative values from peak power (PPW) or GET [5], not the specific ventilatory threshold (VT) values. Furthermore, V̇O_2_ kinetic studies usually start at zero workloads [12,13], which does not reflect real-life scenarios [14]. There is a lack of evidence for V̇O_2_ kinetics during workload transmission from recovery to VT workload.

It is highlighted that the V̇O_2_ slow component is intimately related to intramuscular processes and fatigue development [2], making it a potential marker for monitoring acute and long-term endurance training adaptions. The use of mobile gas analysers in the practical training process has many methodological and practical limitations, and the cost of those systems is relatively high. Near-infrared spectroscopy (NIRS), using devices such as Moxy sensors, is able to directly measure the oxygenation of haemoglobin (Hb) in blood vessels (mainly in capillaries) and myoglobin (Mb) in the muscle cytoplasm, noninvasively indicating skeletal muscle oxidative metabolism in vivo during exercise [15]. Kawaguchi et al. [16] pointed out that peripheral muscle oxygenation reflects the systemic V̇O_2_. Thus, if there is good agreement between oxygen uptake and oxygen desaturation kinetics, the mobile NIRS devices would be valuable tools for practitioners (i.e., coaches) to monitor athletes’ endurance performance and adaptation to training loads in the field at a low cost. Nevertheless, NIRS devices are not without limitations, which need to be addressed, such as adipose tissue thickness and other methodological considerations [17]. However, many studies comparing oxygen kinetics have concentrated on the fast component during moderate intensity in nonsporting or recreational populations. The primary component of deoxygenated Hb (HHb) signal was found to be shorter than in pulmonary V̇O_2_ [18,19,20], and during knee-extension exercise, good agreement was observed between the time characteristics of primary-phase HHb and V̇O_2_ kinetics [18]. Studies focused on cycling exercises have found only moderate correlations between primary-phase kinetic characteristics of muscle deoxygenation and V̇O_2_ at moderate exercise intensity [21]. Some studies have reported no associations [21], especially during heavy-intensity CWR cycling [22]. Similar studies carried out on trained endurance athletes in the heavy-intensity cycling exercise domain, where the slow component is rising, are scarce. Although some fundamental studies point out that approximately 80% of the slow component’s putative mediators come from the contracting muscle [2], and slow component is a significant factor for fatigue development during intense aerobic exercise [6]. Thus, studies describing the associations between the slow components of muscle oxygenation and V̇O_2_ are needed. 

Most studies conducted with NIRS sensors use HHb signals for modelling the kinetics of muscle deoxygenation [17]. HHb signal describes the amount of deoxygenated Hb and is also sensitive to the changes in total Hb; therefore, it is considered to more broadly reflect the metabolic perturbations in local muscle tissue [15]. At the same time, the majority of commercial muscle NIRS sensors predominantly measure muscle oxygen saturation (SmO_2_) values (the relative amount of oxygenated Hb from total Hb) [15], the kinetics of which may not be similar to HHb dynamics. Hence, from a practical point of view, studies about the kinetics of SmO_2_ signals in comparison with V̇O_2_ are needed.

Measuring oxygen uptake through gas exchange parameters represents whole-body O_2_ consumption. Nevertheless, most of the studies conducted with NIRS have used either the left or right limb [23,24] or only the dominant limb [25], without solid arguments. It has been found that NIRS measurements have remarkable intermuscular heterogeneity in deoxygenation kinetics [26], which reduces after priming exercise [27], but less is known about the bilateral homogeneity of the same muscle group and region during workload transitions on bilaterally equal movements such as cycling. Research in the field of cycling biomechanics points to remarkable bilateral asymmetries in the pedalling kinetics, kinematics, and muscular activity characteristics [28], and some evidence indicates that application patterns of the pedalling force may have differences between dominant and nondominant leg [29]. The total mechanical power generated by the cyclist is the summed work rate of both lower limbs; thus, in case one leg is contributing more than the other, can there be some differences in oxygen saturation kinetics as well? If so, can taking both signals into account (i.e., aggregating the signals) more adequately reflect the patterns of pulmonary V̇O_2_ kinetics?

The aim of the present study was to compare and analyse the relationships between pulmonary oxygen uptake and vastus lateralis muscle oxygen desaturation kinetics bilaterally measured with Moxy NIRS sensors in well-trained endurance athletes. 

## 2. Materials and Methods

### 2.1. Experimental Approach to the Problem

A between- and within-subjects cross-sectional design was used to compare pulmonary oxygen uptake and bilaterally measured vastus lateralis muscle oxygen desaturation kinetics and analyse the associations between the homonymic variables of oxygen uptake and muscle desaturation kinetics. Eighteen trained athletes visited the laboratory on two consecutive days at the same time of day and under the same conditions during the second half of the competition period between July and September. On the first day, an incremental test was performed to determine the gas exchange threshold (GET), the ventilatory threshold (VT), and V̇O_2max_ levels from pulmonary ventilation characteristics and determine the power values at those intensity levels—P_GET_, P_VT_, and peak power (PPW) accordingly. Using the threshold powers of day one, on the second day of testing, the constant work rate (CWR) protocol was performed (Figure 1). During CWR tests, the pulmonary ventilation characteristics, the left and right vastus lateralis (VL) muscle [17] O_2_ desaturation (DeSmO_2_), and pedalling power were continuously recorded, and the average signal of both legs’ DeSmO_2_ was computed. The kinetics of pulmonary V̇O_2_ and muscle DeSmO_2_ signals during high-intensity (VT level) CWR cycling were compared, and the relationships between homonymic characteristics were studied. 

### 2.2. Study Participants

Eighteen competitive male cyclists and triathletes participated in this study (details are provided in Table 1). They had at least 10 years of cycling training and competition experience. Study participants were classified as trained and well-trained cyclists according to their physiological performance indicators and training status [30]. The dominant leg for all the participants was the right side, according to the ball-kicking preference. All the participants provided written informed consent. Tallinn Medical Research Ethics Committee approved the study in compliance with the Declaration of Helsinki.

### 2.3. Procedures

#### 2.3.1. Determination of the Gas Exchange Threshold, Ventilatory Threshold, V̇O_2max_, and Peak Power Using Incremental Cycling Test

The athletes underwent an incremental test to volitional exhaustion (cadence decline to 70 rpm) on their own road bike, mounted on a Cyclus2 cycle ergometer (RBM Elektronik-Automation, Leipzig, Germany), considered valid and reliable [31]. After a 7 min warm-up at 90 W, the work rate was increased to 100 W and kept increasing by 30 W every 3 min, and the pedalling cadence was kept constant between 90 and 100 rpm. Throughout the tests, the respiratory and pulmonary gas exchange variables were measured using a breath-by-breath gas analyser (Quark PFTergo, Cosmed, Rome, Italy). Before each test, the gas analyser was calibrated using reference gases (5% CO_2_; 16% O_2_) and a syringe of known volume (3 L), according to the manufacturer’s instructions. V̇O_2max_ was determined as the highest 30 s mean value of V̇O_2_. Athletes’ individual GET, related to the aerobic threshold, was determined as the first breakpoint of V̇CO_2_/V̇O_2_ vs. power relation. The VT related to the anaerobic threshold was identified based on (a) a second slope increase on the curve between minute ventilation (VE) and power, (b) a second increase in ventilatory equivalent for oxygen and ventilatory equivalent for carbon dioxide, and (c) a decrease in the end-tidal partial pressure of carbon dioxide [32]. Two physiologists analysed all data separately, and a third one was involved if there was no consensus. The power values of PPW, P_GET_, and P_VT_ were calculated proportionally [33] according to the time when level criteria (P_GET_ and P_VT_) or end of the test (PPW) were achieved. All the power values were downrounded to the nearest 5 W.

#### 2.3.2. CWR Test

On the second day of laboratory testing, the steady-state protocol was performed (Figure 1) using the same Cyclus2 ergometer and bicycle setup as on the first day. Throughout the exercise protocol, the pulmonary gas exchange variables were measured using a breath-by-breath gas analyser (Quark PFTergo, Cosmed, Rome, Italy), and bilateral pedalling power values were captured with a scientifically validated [34] Favero Assioma Duo pedal power meter (Favero Electronics, srl., Arcade, TV, Italy). The oxygen saturation parameters of the left and right VL muscle were continuously recorded with a mobile NIRS device Moxy Monitor (Fortiori Design LLC, Hutchinson, USA) considered valid and reliable in the whole range of aerobic exercise intensities [24,25]. Moxy Monitor measures the amount of NIR light reaching from one emitter to two detectors (12.5 and 25 mm from the emitter) at four wavelengths (between 630 and 850 nm), with an output sampling rate of 0.5 Hz, and uses the Monte Carlo model for signal processing, which is described in more detail elsewhere [24].

Moxy Monitor probes were symmetrically placed on the left and right VL muscle at two-thirds between the anterior superior iliac and the lateral side of the patella [24]. The sensors were covered with a compatible commercially available light shield and were fixed in place using medical adhesive tape. Before the placement of the probes, body hair was removed, and the skinfold thickness was measured at the site of optode probe placement using a skinfold calliper [17]. The adipose tissue thickness was computed as half of the skinfold measure, and for all subjects, it was less than 10 mm (Table 1).

The Favero Assioma Duo pedal power meter and Moxy Monitor were connected with a Garmin Edge 520 head unit (Garmin, Olathe, KS, USA) using ANT+ signal protocol, and data were synchronously captured, with a frequency of 1 Hz. The Cyclus2 ergometer was controlled using gas analyser software Cosmed OMNIA 2.0 (Cosmed, Rome, Italy),, and the data collected using the gas analyser and Garmin Edge head unit were synchronised.

The CWR test started with a 7 min warm-up at baseline intensity (P_BL_) 150 W or 50% of P_VT_ (for athletes with P_VT_ lower than 300 W), followed by a 10 min priming CWR effort at P_GET_ to mobilise metabolic processes in the cardiovascular system and in working muscles but not to cause remarkable muscle fatigue and blood lactate increase [35]. The test continued with 3 min recovery at P_BL_ followed by 10 min CWR at P_VT_. The baseline and CWR intensities were chosen to mimic the situation of extensive aerobic interval training at the anaerobic level [14]. After 5 min recovery at P_BL_, the maximal severe-intensity CWR effort at the PPW level was performed until the athlete could not hold predefined power. The athletes were verbally encouraged to achieve and hold the V̇O_2max_ level as long as possible to volitional exhaustion. The PPW effort and at least 60 s of passive recovery (until the SmO_2_ achieved peak value) after that were used to measure the physiological amplitude of muscle O_2_ desaturation (DeSmO_2_) for signal normalisation purposes between subjects and contralateral legs for further analysis [36]. The overview of the power values used in the CWR testing protocol is presented in Table 2.

### 2.4. Data Analysis

#### 2.4.1. Signal Processing

Individual breath-by-breath V̇O_2_ data were edited by removing the outlier data that laid 3 SDs from the local mean around the 7 samples’ moving window and linearly interpolated to the frequency of 1 Hz. The time axis of V̇O_2_ and muscle O_2_ saturation (SmO_2_) data were aligned such that time “0” represented the onset of the workload transition from P_BL_ to P_VT_. The SmO_2_ data were collected in the original scale (_orig_SmO_2_) from 0% to 100%, provided by the manufacturer and described in detail by Feldmann et al. [24]. For the purpose of the same directional comparison with V̇O_2_ signal, the muscle O_2_ desaturation value in the original scale (_orig_DeSmO_2_) for the left and right VL was computed by reversing the _orig_SmO_2_ signal (Equation (1)).
(1)$${\;}_{\mathrm{orig}}{\mathrm{DeSmO}}_{2}=100\;-{\;}_{\mathrm{orig}}{\mathrm{SmO}}_{2}\;\left(\%\right)$$

Additionally, for the left and right _orig_DeSmO_2_ signals, the average signal of both legs was also computed. To reduce the influence of the differences between subjects and body sides in terms of anthropometry and sensor placement, all three DeSmO_2_ signals of each participant were normalised against the individual functional range and evaluated during and after maximal cycling exercise at PPW, where the minimum (_orig_DeSmO_2min_) and maximum (_orig_DeSmO_2max_) values of desaturation were captured (Figure 2). The following equation was used to compute the left (_left_DeSmO_2_), right (_right_DeSmO_2_), and average (_avr_DeSmO_2_) VL muscle DeSmO_2_ signal values in percentages (%) for every time (t) sample (Equation (2)) [36]: (2)$$\begin{array}{c} {\mathrm{DeSmO}}_{2}={(}_{\mathrm{orig}}{\mathrm{DeSmO}}_{2}\left(t\right)\;-{\;}_{\mathrm{orig}}{\mathrm{DeSmO}}_{2\min})\\ \;\;\;\;\;\;\;\;\;\;\;\;\;\;\;\;\;\;\;\;\;\;\;\;\;\;\;\;\;\;\;\;\;\;\;\;\;\;\;\;\;\;\;\;\;\;\;\;\;\;\;\;÷{(}_{\mathrm{orig}}{\mathrm{DeSmO}}_{2\max}\;-{\;}_{\mathrm{orig}}{\mathrm{DeSmO}}_{2\min})\times 100 \end{array}$$

#### 2.4.2. Modelling of Oxygen Consumption and Muscle O_2_ Desaturation Kinetics

The body-mass-corrected relative V̇O_2_ (mL/kg/min) (to reduce the effect of between subjects’ body mass differences), _left_DeSmO_2_, _right_DeSmO_2_, and _avr_DeSmO_2_ signals were entered into the kinetic modelling process. The signal processing and modelling of V̇O_2_ and DeSmO_2_ kinetics were performed using validated R-language-based free and open source software VO_2_FITTING (ver. 3) [37].

The V̇O_2_ signal data for the first 15 s after the onset of the exercise were eliminated from modelling to avoid the effect of the cardiodynamic component on fast kinetic analysis. For DeSmO_2_ signals, to improve the preciseness of fast kinetics and TD_p_ computation, the data for the first 10 s were not included in modelling to avoid the potential effect of the initial signal undershoot caused by a rapid increase in blood flow, which is reported in previous studies [21]. For all four signals of each cyclist, the on-transient was modelled with mono- and bi-exponential models (Equations (3) and (4)), characterising the response of V̇O_2_ and VL DeSmO_2_ during 10 min (t = 600 s) CWR exercise [37] as follows:(3)$$Y\left(t\right)=A_{\mathrm{BL}}+H\left(t-{\mathrm{TD}}_{p}\right)\times A_{p}\left(1-e^{-\left(t-{\mathrm{TD}}_{p}\right)/\tau_{p}}\right)$$
(4)$$\begin{array}{c} Y\left(t\right)=A_{\mathrm{BL}}+H\left(t-{\mathrm{TD}}_{p}\right)\times A_{p}\left(1-e^{-\frac{t-{\mathrm{TD}}_{p}}{\tau_{p}}}\right)+H\left(t-{\mathrm{TD}}_{\mathrm{sc}}\right)\times \\ A_{\mathrm{sc}}\left(1-e^{-\left(t-{\mathrm{TD}}_{\mathrm{sc}}\right)/\tau_{\mathrm{sc}}}\right) \end{array}$$
where Y(t) represents the V̇O_2_ and DeSmO_2_ value at the time t, and A_BL_ is the average 60 s signal value (mL/kg/min or %) at cycling in P_BL_ workload. A_p_ and A_sc_ (mL/kg/min or %), TD_p_ and TD_sc_ (s), and τ_p_ and τ_sc_ (s) are, respectively, the amplitudes, the corresponding time delays, and time constants of the fast and slow components of V̇O_2_ and DeSmO_2_ signals (Figure 3 and Figure 4). H represents the Heaviside step function (Equation (5)) [37].
(5)$$H\left(t\right)=\left\{\begin{array}{c} 0,\;t<0\\ 1,\;t\geq 0\; \end{array}\right.$$

To describe the overall response of fast kinetics from the start of the transition, the mean response time (MRT (s)) was computed for all signals as a sum of TD_sc_ and τ_p_. The last modelled value for all four signals and for both models was at the time moment of 600 s (t_600_) when the CWR transition ended, and this characteristic was marked as the EndFit value. The full-response amplitude of the signal was calculated as the difference between EndFit and baseline value (Equation (6)), and for mono-exponential models, the A_end_ was equal to A_p_.
(6)$$A_{\mathrm{end}}=\mathrm{EndFit}-A_{\mathrm{BL}}$$

Since the asymptotic value of the slow component was not necessarily reached at the end of the exercise, the amplitude of the A_sc_ at the end (t_600_) of the test was also calculated, and only the end-exercise slow component amplitude (A_sc_end_) was used for future data analyses (Equation (7)).
(7)$$A_{\mathrm{sc}\_\mathrm{end}}=A_{\mathrm{sc}}\left(1-e^{-\left(t_{600}-{\mathrm{TD}}_{\mathrm{sc}}\right)/\tau_{\mathrm{sc}}}\right)\;$$

The best-fit parameters for mono- and bi-exponential models were chosen using the software to minimise the sum of the squared differences between the fitted function and the observed response expressed as the standard error of regression (SE_regr_). The measure of SE_regr_ was also used to describe the signal variability. To compare the mono- and bi-exponential models, an ANOVA F-test (with the respective residuals’ sum of the squared differences between both models) was applied [37]. If the sum of the squared residuals was statistically significantly lower (*p* < 0.05) in the case of the bi-exponential model, then the characteristics of both fast and slow kinetics were chosen for future analyses; otherwise, only the characteristics of the mono-exponential model were used, which was found in 5 DeSmO_2_ signals of 3 subjects and is presented in the last column of Table 1.

To provide comparable values of relative-response amplitudes of fast and slow kinetics and signal variability for V̇O_2_ and VL muscle DeSmO_2_ signals, the normalisation for all four signals was achieved according to the following Equation (8) [36]: (8)$${\;}_{\;n}Y=\left(Y\left(t\right)-A_{\mathrm{BL}}\right)÷\left(\mathrm{EndFit}-A_{\mathrm{BL}}\right)\times 100$$
where _n_Y is the normalised signal value of _n_V̇O_2_ or _n_DeSmO_2_ signals (%), and Y(t) is the signal value of V̇O2 or DeSmO_2_ at time (t). The normalised signals were modelled according to the functions described previously (Equations (3)–(5)), but A_BL_ was set to 0. Additionally, all time values were fixed as they were in the original models, and only amplitude values were allowed to vary. Normalised primary (_n_A_p_) and slow (_n_As_c_) component amplitudes and standard errors of regression (_n_SE_regr_) were incorporated into future analyses. Normalised signals were also used to average the signals of all the participants for qualitative comparison.

#### 2.4.3. Statistical Analysis

Statistical analyses were performed in SPSS 25.0 (IBM Corporation, Armonk, NY, USA), the data file with individual results for all measured characteristics are available as Appendix A. All data are presented as means ± SDs. To control the assumptions for the parametric tests, the normality and bivariate normality values were controlled using the Shapiro–Wilk test. The homonymous parameters of pulmonary V̇O_2_ and muscle DeSmO_2_ values were compared using a paired Student’s *t*-test or a nonparametric Wilcoxon test if the normality criteria were not achieved. Cohen’s effect size measure (*d*) was used to describe the rate of difference between the evaluated characteristics. The linear associations between the homonymous parameters of pulmonary V̇O_2_ and muscle DeSmO_2_ values were analysed with Pearson or Spearman correlation and simple linear regression methods. The statistical significance was set at *p* ≤ 0.05 for all tests, and Cohen’s *d* > 0.2 (nontrivial effect) was additionally used to evaluate the differences in kinetic characteristics.

## 3. Results

### 3.1. Comparison of Oxygen Consumption and Oxygen Desaturation Kinetics

Table 3 presents the descriptive values of V̇O_2_ and DeSmO_2_ kinetics measured at the work-rate transition from recovery intensity to VT-level heavy-intensity constant-load cycling. The V̇O_2_ and DeSmO_2_ kinetics were compared between the time characteristics and normalised amplitude values. 

All time-based characteristics (τ_p_, TD_p_, and MRT; τ_p_ of V̇O_2_ compared with MRT of DeSmO_2_) describing the initial reaction to increased work rate were significantly shorter in DeSmO_2_ kinetics compared with V̇O_2_ values (*d* > 0.7), but no differences were found between all three DeSmO_2_ signals. The TD_sc_ (accounted only for those subjects who had a slow component in the DeSmO_2_ signal) of the averaged VL DeSmO_2_ signal was significantly shorter than in V̇O_2_ and the right VL DeSmO_2_ kinetics, but no statistical differences were observed between TD_sc_ of V̇O_2_ and the DeSmO_2_ kinetics of the left and right VL signals (Table 3).

The relative primary and slow-component amplitude values showed no differences between V̇O_2_ and DeSmO_2_ (Table 3). No differences were found between DeSmO_2_ signals in terms of absolute and normalised amplitude values. Interestingly, the SE_regr_ values that describe modelling accuracy and signal variability were significantly (*p* = 0.03) lower in the left VL DeSmO_2_ signal (1.42 ± 0.70%) than in the right side (1.75 ± 0.63%). At the same time, the variability of the averaged DeSmO_2_ signal (SE_regr_, 1.24 ± 0.46%) was even smaller, and this difference was clearly significant (*p* < 0.01) compared with the right- and left-leg values. The same dynamics between DeSmO_2_ parameters were present on the normalised standard error of regression (_n_SE_regr_), and because of the normalisation, V̇O_2_ and DeSmO_2_ data were comparable, and there was also a significant difference between V̇O_2_ and all DeSmO_2_ signals. 

### 3.2. Associations between Oxygen Consumption and Oxygen Desaturation Kinetics 

In Table 4, the associations between the equivalent characteristics of V̇O_2_ and DeSmO_2_ kinetics are provided. At the baseline (BL), there was no correlation between V̇O_2_ and any of the three VL DeSmO_2_ signals. The primary amplitude values of V̇O_2_ and all the DeSmO_2_ signal kinetics were significantly and positively correlated. The time values describing the fast component (τ_p_, TD_p_, and MRT) of V̇O_2_ and VL DeSmO_2_ kinetics were not significantly correlated. The correlation was also absent in A_sc_, EndFit, and normalised slow-component amplitude (_n_A_sc_) data. However, there was a positive correlation between the normalised primary amplitude of V̇O_2_ and the averaged VL DeSmO_2_ kinetics. The overall response value to the workload transmission of V̇O_2_ (A_end_) was positively correlated with all three DeSmO_2_ homonymic values. There was a significant (*p* < 0.01) positive correlation between TD_sc_ values of V̇O_2_ and right VL DeSmO_2_ and also the averaged VL DeSmO_2_ signal. 

Table 4 shows that the kinetics of the left and right VL DeSmO_2_ were not very well correlated at the interpersonal level, and only the baseline and primary amplitude values of contralateral VL DeSmO_2_ kinetics were significantly correlated. 

## 4. Discussion

To our knowledge, this is the first investigation to compare and study the relationships between pulmonary oxygen uptake and bilaterally measured vastus lateralis muscle’s oxygen desaturation kinetics in the trained endurance athlete population during workload transition that mimics traditional AL exercise after priming at P_GET_. The analysed exercise work rate was on the border of heavy and severe intensity, which should induce a slow component in V̇O_2_ and DeSmO_2_ kinetics [5,6], but the majority of previous studies about the relationships between V̇O_2_ and muscle oxygenation kinetics have been focused on the primary-phase kinetics [4,5,10]. Most studies with NIRS sensors have traditionally measured only one body side without clear reasoning [23,24]. Even when the same muscles are measured bilaterally, only one side is reported [38]. The additional aim of the present study was to evaluate the homogeneity of DeSmO_2_ kinetics of the same region of contralateral VL muscles.

### 4.1. Fast Component

Oxygen kinetics during the exercise has three phases; phases one and two indicate the fast component of the V̇O_2_ kinetics. A comparison of oxygen consumption and muscle oxygen desaturation kinetics mainly revealed differences in time-based parameters (τ_p_, TD_p_, and MRT) describing the initial reaction to increased work rate being significantly shorter and faster in DeSmO_2_ kinetics than in V̇O_2_ kinetics (Figure 5), and this is well supported with previous findings [18,19,20]. Compared with the literature [19,39,40], in our study, phase one (TD_p_) was considerably shorter in V̇O_2_ kinetics, approximately 10 s, compared with the previously reported approximately 15–20 s in duration. The values of the primary time constant in our study were in the range reported to be common for trained endurance athletes during workload increase to heavy-intensity level cycling [41,42]. The shorter TD_p_ in the present study may be related to the higher baseline level compared with the more frequently used zero workloads before the effort, which is also explainable by the priming effect that is shown to shorten the TD_p_ [43]. Similar TD_p_ values, as in our study, are reported by Murias et al. [20]. Shorter and faster primary kinetics can also be described by Jones et al. [44] findings, who showed that faster oxygen kinetics is positively associated with higher aerobic capabilities. 

Similar to V̇O_2_ kinetics, the TD_p_ values of DeSmO_2_ were shorter than those reported in most earlier studies managing moderate workload transitions without priming [18,19,20,26]. Studies in which heavy-intensity CWR cycling exercises are used have demonstrated more similar TD_p_ values [26,40], as presented in the current study for VL deoxygenation kinetics. In combination with longer TD_p_ values, the earlier studies also presented faster τ_p_ values for muscle deoxygenation during moderate-intensity cycling without priming and from low workload transition [18,19,20,26]. At the same time, the research of Saitoh et al. [27] demonstrated that priming conditions significantly shorten the TDp and increase the time constant of fast kinetics of the HHb signal. Additionally, the higher exercise intensity has also been shown to shorten the TD_p_ but without changes in the time constant [40]. 

Compared with most previous studies, which have modelled the HHb signal, we used the muscle oxygen saturation signal (DeSmO_2_ = 100 − SmO_2_) for kinetic modelling, which is more commonly displayed by commercially available wearable NIRS sensors [15]. Our experiences demonstrated that the DeSmO_2_ signal presented a good fit to exponential fast kinetic modelling for all subjects, and for the majority of cases, we did not detect signal undershoot immediately after the onset of exercise, as described in previous studies using the HHb signal [18,19,20,26,27,40,45]. The HHb signal is sensitive to the changes in the total Hb in the captured region of the muscle [15], and it has been demonstrated that at the onset of exercise, at least in conditions with low baseline and without priming, the signal of the total Hb reduces and after several seconds, starts to rise again [19,45]. The SmO_2_ signal is not so sensitive to fast changes in the total Hb and represents the relative distribution of oxy and deoxy Hb. The prior priming exercise and the elevated baseline have also been shown to reduce fast hemodynamical reactions during the start of workload increase [15] and make signal kinetics more stable [27], which can be the main reasons for the controversial results of our study compared with the majority of previously published data.

Some previous studies have reported good or moderate agreement between the homonymous time characteristics of primary-phase kinetics between V̇O_2_ and muscle deoxygenation at moderate intensity [18,22]. Nevertheless, there were no correlations between the fast-component time-based parameters of V̇O_2_ and DeSmO_2_ kinetics in any aspect of our study. This result supports similar findings of some previous results also declaring a lack of association between VL muscle deoxygenation and pulmonary V̇O_2_ kinetics during moderate intensity [19,20,45], but especially during heavy-intensity domain CWR exercises [22,26,27]. The reason for this outcome may be that at the beginning of the effort, there is a high number of different mechanisms influencing the oxygen kinetics, such as rapidly elevated breathing, increased work level of stabilising muscles, etc. Therefore, interpersonal responses are highly variable [15]. Similarly, the inconsistency of the breathing pattern at the start of the exercise may lead to measurement errors in V̇O_2_ actual signal, thus masking the results we saw in our dataset, mainly because there was only a single effort the participants performed. Analogously, Koga et al. [26] demonstrated the spatial heterogeneity of the quadriceps muscle and, therefore, a lack of correlation between the muscle and pulmonary oxygen uptake.

Nevertheless, during the heavy load, the primary component’s time-based parameters were more homogeneous than those at moderate intensity [26]. In addition, the priming exercise has been shown to reduce spatial heterogeneity in deoxygenation kinetics [27]. Additionally, because oxygen uptake is measured using the breath-by-breath method, but the muscle desaturation signal’s frequency was captured at 1 Hz, synchronising those signals may lead to a 1 s error margin between time series. This may influence the correlations between time variables lasting less than 30 s.

The comparison of primary-phase normalised amplitude (_n_A_p_) values did not show any differences in the relative proportions of V̇O_2_ and all three studied DeSmO_2_ signals in the dataset. Even though the proportions of normalised primary amplitude signals were not significantly different, there was a correlation only between V̇O_2_ and the average DeSmO_2_ signal. This may be because normalised values represent more the spectrum of this unique test, not the full potential of one’s physiology. However, the primary amplitude (A_p_) values of V̇O_2_ and all the DeSmO_2_ signal kinetics were significantly and positively correlated. These values allow an understanding of the full potential and response of individual physiology to the heavy cycling load, indicating that the first response in the muscle O_2_ saturation signal of the leading power-producing muscle group may adequately describe the general metabolic perturbation at the start of heavy-intensity cycling exercise. 

### 4.2. Slow Component

The third component of oxygen kinetics is described as a continuous rise in oxygen uptake and is defined as a slow component. In this study’s dataset, not all the subjects had slow components detected in the VL DeSmO_2_ kinetics of one (n = 2) or both legs (n = 1) (Table 1), and these kinds of cases have been reported in the scientific literature [22]. Therefore, only those subjects who had named components were accounted for in the analysis of slow-component-related data. In the characteristics describing the absolute or relative amplitude of the slow component, there were neither any differences nor associations between V̇O_2_ and DeSmO_2_ kinetics, supporting the earlier findings [22]. Contrasting findings were found from the timing characteristics of the slow component. The TD_sc_ of V̇O_2_ kinetics was an average of 140.3 ± 56.0 in our study, and this was very similar to the values (258 ± 98 s) reported by Bearden and Moffatt [39] measured during a cycling exercise with comparable conditions (priming and raised baseline), but with untrained males. The latter study also demonstrated that priming exercises significantly (~40 s) increase the duration of TD_sc_ [39]. Our findings also match well with another study [22], in which V̇O_2_ TD_sc_ values were found to be 139.1 ± 8.1 s, measured in traditional heavy-intensity CWR transition from zero baselines without priming and with untrained subjects. Nevertheless, the TD_sc_ values of the deoxygenation signal were significantly shorter (75.0 ± 14.0 s) than those measured in the current study. One reason for this can be related to the priming effect that, similarly to the pulmonary V̇O_2_ kinetics, possibly delays the slow component of muscle deoxygenation [39]. Secondly, the endurance training history is associated with a smaller and later increase in the slow component [2]. The third reason can arise from the specifications of modelled DeSmO_2_ signal, instead of the traditionally used HHb signal; muscle O_2_ saturation does not account for the change in the total Hb, and the slow component rise in HHb can be started by an increase in the total Hb level, caused by elevated blood flow, [15] instead of an increased proportion of desaturation. In accordance with [46], in the present study, the TD_sc_ of the aggregated signal of right and left VL DeSmO_2_ was significantly shorter than V̇O_2_ TD_sc,_ (Figure 5), and the right leg’s signal was longer than the average of both legs’ DeSmO_2_. Because the coefficient of the variation for the average signal was significantly lower, the latter probably indicates the cooperation left and right legs for sharing the load and reflecting general oxygen uptake in a body (this topic is discussed in detail below). The methodical reason for why the average TD_sc_ value is lower than the separate values for the left and right legs can be explained by understanding that it is not a mathematical average of discrete numbers. Instead, the signals are aggregated beforehand, and then the kinetics are modelled again. This means that in the averaged signal, the slow component represents the earliest start of the slow component, either from the left or right side (visually illustrated in Figure 5).

The substantive importance of the earlier start of the slow component in the muscle desaturation signal (started earlier from either the left or right side) points to the intramuscular mechanisms inducing additional O_2_ increase in pulmonary gas exchange. This mechanistic relationship is also supported by our other finding about strong correlations (r > 0.8) between the TD_sc_ of V̇O_2_ and the averaged (as well as the right leg’s) DeSmO_2_ signals. In this aspect, our study supports the findings of Kawaguchi et al. [16].

During heavy load intensity, after the primary component, there is a steady-state phase during which the body adapts to the load, but thereafter, the lack of resources develops into a slow component as a continuous rise in the global oxygen uptake, and this has been related to different global and local mechanisms [2,7]. It is stated that the largest value of the slow component is related to the intramuscular component [2,8,12], and our results about the timing of the slow component clearly indicate that those processes start at the local muscle level. Additionally, the relative values of slow-component amplitudes from the end-exercise response did not differ between V̇O_2_ and DeSmO_2_ signals. This indicates that in most cases (in our sample, there were some cases without a slow component in the DeSmO_2_ signal either in one or both legs), the VL muscle O_2_ saturation reflects the additional metabolic needs at the intramuscular level during heavy cycling exercise quite well. The main intramuscular components leading to the increased O_2_ needs are the progressive recruitment of type II muscle fibre types, the increase in the muscle’s inner temperature, and proton leak through the inner mitochondrial membrane [7]. The factors that can additionally increase global pulmonary O_2_ consumption are related to increased cardiorespiratory work and elevated levels of postural stabilising muscles activity [47], as well as the compensatory increase in synergistic muscles and changes in the pedalling technique [8]. It can be hypothesised that the relative number of different mechanisms for slow-component development has large interindividual variability. This can explain the lack of correlations between the slow-component amplitude values of V̇O_2_ and DeSmO_2_ kinetics presented in our results. The reasons for the absence of slow components in the DeSmO_2_ kinetics on one side of the body for two cases in our study can be related to the high-intensity workload level and the possible postural control, leg blood flow restriction, or pedalling-power-producing asymmetries. This may induce the use of almost all of the total oxidative capacity of one side’s VL already from the start of the exercise (A_p_ of DeSmO_2_ was more than 98% for both cases), in which case additional pulmonary O_2_ increase will be generated by other working muscle groups and by the contralateral side. The subject without a slow component in both legs’ DeSmO_2_ had relatively low A_p_ and A_end_ (less than 80%) values. It can be hypothesised that the particular pedalling technique of that subject was oriented for higher usage of other muscles/muscle groups responsible for additional pulmonary O_2_ increase.

### 4.3. The General Response to the Exercise and Bilateral Differences in Oxygen Desaturation

The sum of the primary- and slow-component amplitude, defined as end-exercise response amplitude (A_end_), had a significant positive association with all the counterparts in both cases except between the left and right VL DeSmO_2_ signals. We observed a similar pattern in the primary amplitude as well. Regardless of the low agreement between the slow-component amplitudes of V̇O_2_ and DeSmO_2_ kinetics, discussed in detail in the previous section, the end-exercise response of DeSmO_2_ kinetics reflects the general metabolic reaction to the exercise relatively adequately.

The standard error of regression (SE_regr_) values that describe modelling accuracy and signal variability were lower in the left VL DeSmO_2_ signal than in the right side, and furthermore, the variability of the averaged DeSmO_2_ signal was even smaller, and this difference was clearly significant compared with the right and left leg’s values. If the left and right legs would contribute to the process separately, then the variation in the separate values for legs would carry forward to the aggregated value as well, but this was not the case. Therefore, the synergic bilateral metabolic sharing during high-intensity CWR cycling may be hypothesised. Nevertheless, future studies about the synergy phenomenon, commonly described in motor control processes [48] at the metabolic level are needed. 

Our study demonstrated the relatively low agreement between the VL DeSmO_2_ characteristics of contralateral legs_,_ and only significant correlations between the left and right legs were found in A_BL_ and A_p_ values, but the temporal and slow-component response characteristics were insignificantly associated. The bilateral comparison of DeSmO_2_ kinetics did not demonstrate direction-/dominance-directed differences in time- or amplitude-based characteristics. As mentioned above, there was only one direction-related significant difference, namely the signal variability on the right/dominant side was higher than that on the left. Some evidence in the cycling literature [28,29] demonstrates that the dominant leg generates power more effectively over the pedalling cycle, which is coordinatively more challenging, and therefore the work performed by a single muscle group may be more variable. By contrast, the force production in the left leg is more impulsive and stable during the pushing phase of the pedalling cycle [28,29]. Future studies are needed to investigate possible associations between metabolic and motoric processes in working muscles during heavy-intensity CWR cycling.

As stated above, to our knowledge, there have not been previous studies using similar methods to account for the differences between the leg’s oxygen desaturation kinetics. Nevertheless, our data suggest the importance of accounting for both limbs’ data together because there seems to be no synchronisation of desaturation kinetics between legs. On the one hand, this demonstrates the existence of remarkable asymmetry in muscle oxygenation or oxygen saturation, which can be caused by other anatomical or neuromuscular inequalities. Nevertheless, our results also demonstrate that those asymmetries are mainly unrelated to leg dominance. Our results also point to the bilateral synergistic sharing of metabolic load; mostly, the characteristics of the aggregated deoxygenation signal were more strongly correlated with the pulmonary oxygen uptake kinetics than with contralateral VL muscles separately. In particular, the values for the left leg, as well as the nondominant leg, had a lower or nonsignificant association with general oxygen kinetics. Therefore, our results indicate that measuring both limbs or at least preferring the dominant leg over the nondominant leg gives more accurate information about general oxygen utilisation.

### 4.4. Practical Applications

In general, VL is a valid muscle for describing the dynamics of metabolic processes during high-intensity cycling. Our results demonstrate that, in most cases, the existence of the slow component of oxygen consumption can be detected from the NIRS signal, and the relative response amplitudes of primary and slow components are similar to pulmonary V̇O_2_ values. This indicates that during endurance exercises at the anaerobic threshold level, NIRS devices (Moxy Monitor in the present case) can be valuable tools for monitoring internal metabolic alterations at the local muscular level, as indicated in the recent reviews by Perrey [17,49]. Moreover, using the NIRS device on the dominant limb is recommended if only one sensor is present. Otherwise, measuring both limbs and aggregating the data more precisely reflects the metabolic processes during high-intensity cycling. 

## 5. Conclusions

The relative response amplitudes of the primary and slow components of VL desaturation and pulmonary oxygen uptake kinetics did not differ, and the primary amplitude of muscle desaturation kinetics was strongly associated with the initial response rate of oxygen uptake. Compared with pulmonary O_2_ kinetics, the primary response time of the muscle desaturation kinetics was shorter, and the slow component started earlier. There was good agreement between the time delays of the slow components describing global and local metabolic processes. No differences caused by leg dominance (body side) were found in muscle desaturation kinetics. However, there was a low level of agreement between contralateral desaturation kinetic variables. The averaged DeSmO_2_ signal of the two body sides represented the oxygen kinetics more precisely than the right- or left-leg signals separately.

## Figures and Tables

**Figure 1 jfmk-08-00064-f001:**
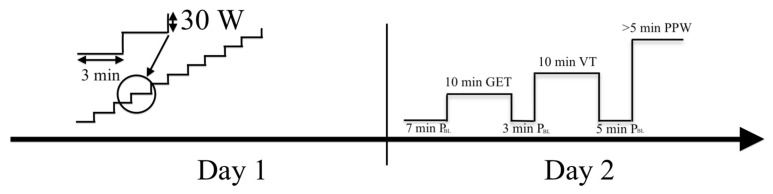
Timeline of the study: P_BL_ (baseline); GET (gas exchange threshold); VT (ventilatory threshold); PPW (peak power).

**Figure 2 jfmk-08-00064-f002:**
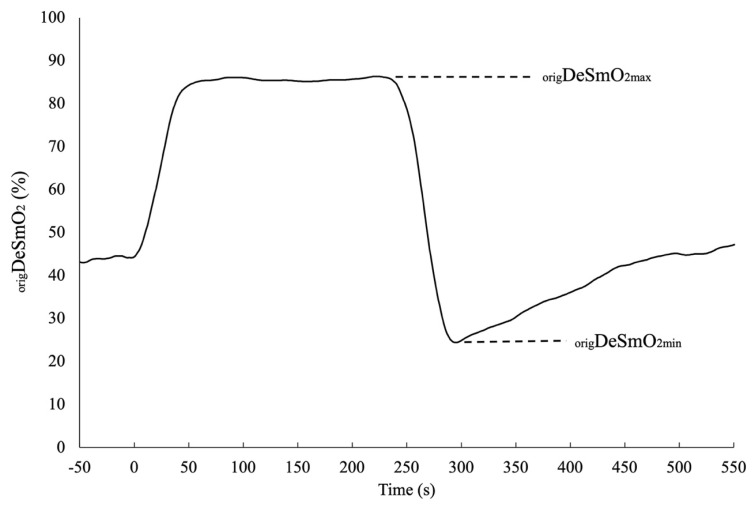
The typical functional amplitude of the DeSmO_2_ signal in the original scale for one participant during and after CWR cycling at PPW intensity.

**Figure 3 jfmk-08-00064-f003:**
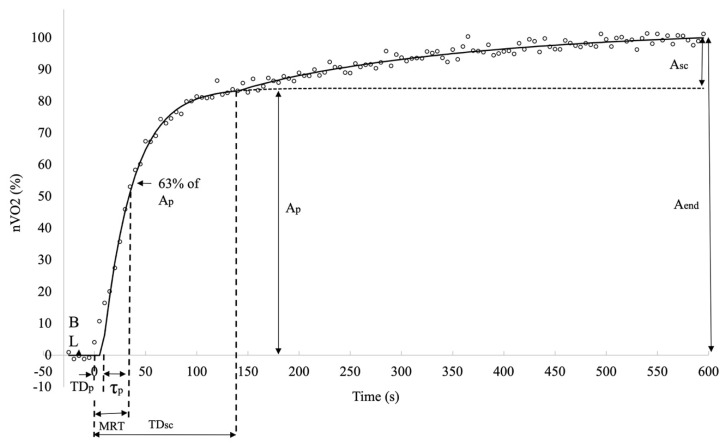
Descriptive diagram of measured characteristics of oxygen consumption kinetics in averaged V̇O_2_ signals (white circles) of all study participants during cycling at VT level power.

**Figure 4 jfmk-08-00064-f004:**
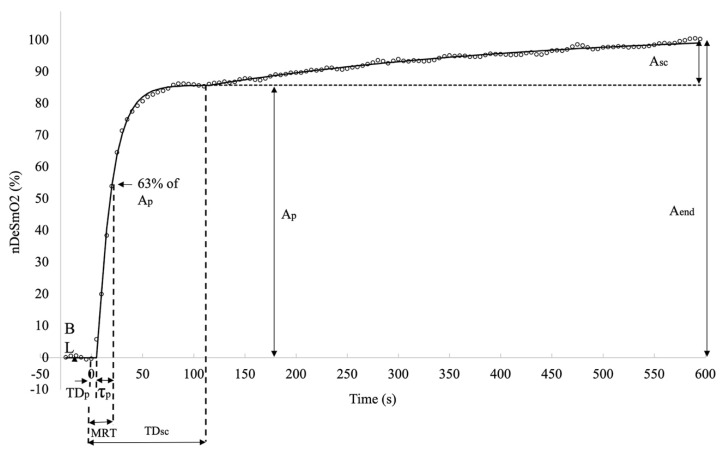
Descriptive diagram of measured characteristics of muscle oxygen desaturation kinetics in averaged right VL DeSmO_2_ signals (white circles) of all study participants during cycling at VT level power.

**Figure 5 jfmk-08-00064-f005:**
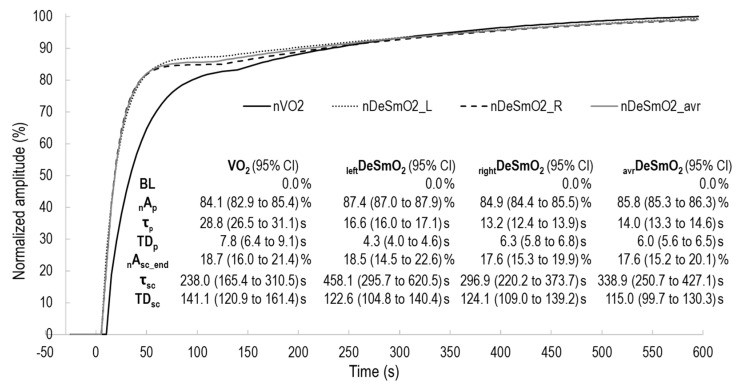
The two exponentially modelled kinetics of V̇O_2_ and DeSmO_2_ signals of averaged time course values of all participants; characteristics of kinetics are presented with 95% confidence intervals (CIs).

**Table 1 jfmk-08-00064-t001:** Descriptive data of study participants.

Subject	Age (y)	Height (m)	Weight (kg)	Half of Skin-Fold (mm)	Yearly Training Hours	Yearly Cycling Distance (km)	PPW (W/kg)	VO_2max_ (mL/min/kg)		Duration of Normalisation Trial (s)	Comments
								Incremental Test	Normalisation Trial at PPW		
A	42	1.759	85.75	9	830	14,200	3.97	57.4	59.1	361	
B	20	1.895	82.45	5	466	10,200	4.85	66.7	68.5	240	
C	45	1.777	78.85	8.5	530	6500	3.93	50.6	52.1	121	
D	46	1.841	83.3	7.5	525	9400	4.08	54.2	55.5	299	
E	52	1.814	75.2	6	425	6200	3.72	58.7	59.0	340	L; R; A
F	44	1.77	72.4	4.5	510	7500	5.11	64.0	63.1	306	
G	43	1.798	81.2	5	560	10,900	4.99	71.0	73.0	387	
H	45	1.829	88.4	6	445	8300	4.64	62.2	61.6	330	
I	48	1.841	91.2	8.5	480	9900	4.00	64.0	64.0	355	
J	37	1.845	79.45	7	670	13,400	4.41	62.7	64.6	363	
K	46	1.96	91.6	4	1000	14,900	4.53	64.5	65.1	365	R
L	44	1.849	85.25	6.5	590	10,800	4.22	59.4	59.0	296	L
M	46	1.847	81.7	6	415	9400	4.65	70.7	72.8	295	
N	41	1.913	79.95	3	560	14,400	5.13	74.6	75.8	373	
O	37	1.839	79.6	7.5	415	10,400	4.84	66.3	67.7	335	
P	33	1.876	92.1	9	520	10,700	4.40	60.0	61.2	388	
Q	44	1.849	78.2	5	640	9000	4.73	71.8	70.1	292	
R	50	1.76	76.5	7.5	415	6000	3.86	54.4	56.0	300	
Mean	42.4	1.837	82.4	6.4	555	10,117	4.45	63.0	63.8	319.2	
SD	7.2	0.053	5.7	1.8	154	2744	0.45	6.6	6.6	63.4	

No SC in DeSmO_2_ kinetics: (L) left leg; (R) right leg; (A) average signal.

**Table 2 jfmk-08-00064-t002:** Overview of power values of different intensities used during constant-load exercise protocol and bilateral pedalling power values measured during CWR cycling.

Intensity	P (W)	Pkg (W/kg)	P_LeftPedal_ (W)	P_RigthPedal_ (W)
BL/Recovery	146.1 ± 9.2	1.78 ± 0.13		
Priming at GET	237.5 ± 32.1	2.89 ± 0.39		
CWR at VT	301.9 ± 34.6	3.67 ± 0.38	147.5 ± 19.7	151.2 ± 22.3
∆P of CWR	155.8 ± 27.5	1.89 ± 0.30		
Normalisation at PPW	366.1 ± 40.8	4.45 ± 0.45		

**Table 3 jfmk-08-00064-t003:** The characteristics of original VO_2_ and DeSmo_2_ signals kinetics measured during CWR cycling at VT2 intensity.

	V̇O_2_	_left_DeSmO_2_	_right_DeSmO_2_	_avr_DeSmO_2_
A_BL_ (mL/kg/min or %) #	30.1±2.7	27.8± 10.5	23.2 ± 11.1	25.5 ± 9.6
A_p_ (mL/kg/min or %) #	20.5 ± 3.8	53.7 ± 11.1	53.5 ± 11.7	53.9 ± 9.4
τ_p_ (s)	26.5 ± 4.2	14.2 ± 4.2 *	12.6 ± 3.9 *	13.7 ± 3.4 *
TD_p_ (s)	9.5 ± 3.7	6.1 ± 2.4 *	6.4 ± 2.3 *	6.1 ± 2.0 *
MRT (s)	36.0 ± 5.4	20.3 ± 4.6 *	19.0 ± 3.2 *	19.8 ± 3.4 *
TD_sc_ (s) ¤	140.3 ± 56.0	137.7 ± 79.7	138.8 ± 72.2	122.8 ± 59.6 *^,R^
A_sc_ (mL/kg/min or %) ¤ #	4.1 ± 1.1	9.5 ± 6.1	10.7 ± 5.7	9.3 ± 4.7
EndFit (mL/kg/min or %) #	54.6 ± 5.9	89.9 ± 7.2	86.2 ± 9.5	88.1 ± 6.5
A_end_(mL/kg/min or %) #	24.6 ± 4.1	62.1 ± 10.3	63.1 ± 10.8	62.6 ± 9.0
SE_regr_ (mL/kg/min or %) #	1.62 ± 0.50	1.42 ± 0.70	1.75 ± 0.63 ^L^	1.24 ± 0.46 ^L,R^
_n_A_p_ (%)	83.3 ± 4.2	85.5 ± 10.2	83.6 ± 10.2	85.0 ± 7.7
_n_A_sc_end_ (%) ¤	17.0 ± 4.0	15.1 ± 9.8	16.9 ± 8.7	14.7 ± 7.2
_n_SEr_egr_ (%)	6.72 ± 1.97	2.32 ± 1.15 *	2.85 ± 1.22 *	2.00 ± 0.75 *^L,R^

Notes: *—significantly different from VO_2_ (*p* < 0.05; *d* > 0.2); ^L^—significantly different from left DeSmO_2_ (*p* < 0.05; *d* > 0.2); ^R^—significantly different from right DeSmO_2_ (*p* < 0.05; *d* > 0.2); ¤—counted only when sc was presented; #—no comparison between VO_2_ and DeSmO_2_ values was made.

**Table 4 jfmk-08-00064-t004:** Associations between characteristics of VO_2_ and DeSmO_2_ kinetics.

	V̇O_2_ and	V̇O_2_ and	V̇O_2_ and	_left_DeSmO_2_ and
	_left_DeSmO_2_	_right_DeSmO_2_	_avr_DeSmO_2_	_right_DeSmO_2_
A_BL_ (mL/kg/min or %)	−0.224	−0.121	−0.331	0.559 *
A_p_ (mL/kg/min or %) ¤	0.637 **	0.806 **	0.746 **	0.754 **
τ_p_ (s)	−0.134	0.413	0.036	0.251
TD_p_ (s)	−0.073	0.192	0.189	0.467
MRT (s)	−0.226	−0.233	−0.355	0.180
τ_p_ of V̇O_2_ (s) and MRT of DeSmO_2_	−0.290	−0.067	−0.387	
TD_sc_ (s) ¤	0.263	0.828 **	0.865 **	0.200
A_sc_ (mL/min/kg or %) ¤	0.044	0.262	0.330	0.350
EndFit (mL/min/kg or %)	−0.043	0.294	0.208	0.211
A_end_ (mL/min/kg or %)	0.584 *	0.561 *	0.678 **	0.458
_n_Ap (% of A_end_) ¤	0.330	0.481	0.493 *	0.473
_n_Asc (% of A_end_) ¤	0.163	0.375	0.358	0.416

Notes: *—significant correlation (*p* < 0.05); **—significant correlation (*p* < 0.01); ¤—counted only cases where sc was presented.

## Data Availability

The original data file is available as Appendix A.

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
