# Peer review of "Oxygen Uptake and Bilaterally Measured Vastus Lateralis Muscle Oxygen Desaturation Kinetics in Well-Trained Endurance Cyclists"

_jfmk, 2023, doi:10.3390/jfmk8020064_

Round 1
Reviewer 1 Report
General comments
The present study aimed to was to compare and analyze the relationships between pulmonary oxygen uptake and Vastus Lateralis (VL) muscle oxygen desaturation kinetics measured bilaterally with Moxy NIRS sensors in trained endurance cyclist. This well-written article with an adapted and robust methodology presents data that the literature has already established. Overall, the results of this study are well known. All the points presented in the conclusion have already been brought to light. Therefore, the originality of this study does not appear to be really present. Based on the conclusion of the present study, most findings are already well known (or not in agreement) with the literature, except here that measurements were carried out in well-trained endurance cyclists and that both limbs were measured L495-499
The authors seem to indicate that it is necessary to measure the muscle oxygenation of the 2 lower limbs but conclude in the end that it is more accurate to take the average in assessing the muscle deoxygenation, while they criticized it beforehand. The rationale behind this statement is very vague. What is really the degree of accuracy provided in the kinetics parameters and its real impact? Ultimately, the question of measuring one or two limbs depends on the target sport. Here for cycling, the movements of the lower limbs are cyclic and only the dominant limb could show some little variation. For acyclic sport requiring propulsive limb characteristics there may be value in measuring both sides.
Finally, some results (no correlation between the muscle oxygenation and pulmonary VO2 kinetics) require clarification and explanation; the discussion is sometimes too speculative and not in line with the knowledge of the literature. The methodological limitationss must be addressed, including those inherent to the Moxy device proposing basically SmO2%.
Introduction
L29-83. The first part on oxygen uptake kinetics is too long. It should not be an extensive review of literature on VO2 kinetics but rather presenting some key findings in building your rationale.
L76-80. Please make sure that all physiological indicators and thresholds are useful in this study.
L92-93. Authors should introduce here past studies combining VO2 and deoxygenation kinetics where fast and slow components were compared whatever the mode of the motor task. For instance, concentric work and heavy intensity as encountered in cycling (DOI:10.1111/j.1475-097X.2004.00554.x). In this study the majority of the slow rise of O2 arose from the exercising limbs and the fast components of both signals were correlated (tight coupling but with a delay).
L94-96. Might be relevant to introduce here the systematic review indicating application of NIRS monitoring in sports (DOI: 10.1007/s40279-017-0820-1)
L108-109. Regarding the past comment on that point, authors should propose why slight differences in oxygen saturation kinetics between dominant and nondominant will occur? and how they can more strongly describe the patterns of pulmonary V̇O2 kinetics? Findings in the literature are clear on the coupling between oxygenation kinetics and VO2 kinetics (ie. Faster response for muscle oxygenation, see DOI: 10.1152/japplphysiol.00695.2002, DOI:10.1111/j.1475-097X.2004.00554.x)
L109. Based on previous comments introduce key findings comparing and analysing the relationships between pulmonary oxygen uptake and Vastus Lateralis muscle oxygen desaturation kinetics (see proposed references among other), and build a stronger rationale on them before proposing a hypothesis; this is currently lacking).
Methods
When determining kinetics of either VO2 or muscle saturation, reliability is a key issue to consider due to noise especially in the fast component (see DOI: 10.1152/jappl.1987.62.5.2003) where multiple trials are often required. Confidence intervals of the kinetic parameters estimated from the fitting model should be proposed in any cases.
Figure 1. Please change ‘ by min. Add the name of all abbreviations (BL, GET, VT, PPW) in the legend/title.
L129. Why kinetics of pulmonary VO2 and muscle SmO2 were not compared also at the lowest exercise intensity (GET, moderate domain) and at PPW level (time trial to volitional exhaustion) ?
L129. Correct VO2max by VO2 here.
L150. Please inform the readerships about the calibration procedure for measuring pulmonary VO2.
L172. Does the Moxy monitor is valid and reliable within the context of heavy (and severe) constant work rate exercise?
Please inform on the Moxy device (sampling rate, receiver-emitter sensors, inter optode distance…)
L173. What about the measurement of adipose tissue thickness as a confounding factor? Please refer to guidelines in DOI: 10.1007/s40279-017-0820-1
Replace ‘ by min throughout the text.
Please use only CWR abbreviation (common) and not CWE, CWL.
L226. It is rather strange that the time delay after onset of exercise has been removed arbitrary. The on-transient VO2 kinetics can capture easily this parameter before modelling the fast response (amplitude and time constant). In this case why TDp (time delay for the fast component = duration of the cardiodynamic phase 1) is introduced in equations 3 and 4? Please clarify.
Figures 3 and 4. Progressive increase in muscle oxygenation and VO2 over time look like a linear drift rather than an exponential term. Have you tested this modelling approach? Standard error of regression should be less as well as the sum of squared residuals, likely lower.
L274. Please make sure that parametric tests are applicable for your data set. Normality must to be assessed at least.
Add the effect size for the statistical tests.
L297. Replace Non by No
L306. What about the correlation of time constant and MRT responses between VO2 and oxygen desaturation kinetics? These are the fundamental correlations to test according to the literature.
Discussion
L328. Remarkable term is not suitable, and can be removed.
L340-356. Please be accurate when you are discussing “faster oxygen kinetics”. Indicate clearly when you deal with either the time delay or the time constant (more relevant) and MRT responses.
For TDp please see previous comments in the methods. Something is wrong. Your results should be comparable to other studies.
L352. This absence of correlation appears not possible! See previous comments with provides studies where this relation does exist, all the more muscle O2 kinetics is representing the fast phase of VO2 kinetics (intramuscular mechanism evoking pulmonary VO2 in cycling). The issue can be the SmO2 signal used here as compared to the literature focused on the desoxyhemoglobin signal (HHb). Please adjust accordingly.
Regarding the slow phase, did the authors verify that TD occurs when the fast response was almost achieved (=4 or 5 * time constant of the fast phase). Since time constant for SmO2 faster, TD of the slow phase should arrive earlier.
Reviewer 2 Report
GENERAL COMMENTS
I am familiar with the topic under review and the material presented is off interest. However, I am uncertain if the manuscript really offers and new or novel information that readers would find important. For instance, the finding that primary and slow component oxygen kinetics for pulmonary and local tissue sites were similar does not seem particularly noteworthy. I am certain other papers in the past have reported similar findings. I think it is very important to the authors to offer more compelling reasons why this study was done, and therefore why it should be published. As the manuscript now reads, it is devoid of fatal flaws yet provides little in the way of new and substantive information.
SPECIFIC COMMENTS
Line 23: change “Aswell a” to “As well as”
Line 29: remove the hyphen
Line 115: change “in-subjects” to “within-subjects”
Line 118: change “18” to “Eighteen”
Line 119: change “in” to “under”
Lines 183, 185 & 186: change “ ´ ” to “minutes”
Line 330: omit “operating”
Line 412: change “from” to “at”
Line 414: omit “that”
Line 423: change “hypnotized that” to “hypothesized”
Round 2
Reviewer 1 Report
The authors revised well their manuscript and did reply enough to the previous concerns addressed. Please find below final minor comments for your convenience.
L23. Abstract. The sentence As well as ... is not a sentence. Please correct.
Introduction. HHb to use (not HHB), same for Hb (and not HB)
Lines 147-148. Note that HHb signal is much less sensitive to the changes in total Hb. This is important to consider when using muscle NIRS. HHb being the relevant signals of interest. Please change the sentence accordingly.
Lines 257-265. It should be "10 min" (not necessary to write minutes)
Practical applications
Lines 654-655. ..." can be valuable tools for monitoring internal metabolic alterations at the local muscular level". This idea was exposed (practical applications) recently in 10.3389/fspor.2022.864825 (might be relevant to add it since It fits yours thoughts/results).
Reviewer 2 Report
Line 45: Change “Sprint-type athletes “ to “Sprinters”
Line 46: Change “endurance-type athletes “ to “endurance athletes “
Line 120: insert “not” between “are” and “without “
